# New Pharmacological Strategies against Pancreatic Adenocarcinoma: The Multifunctional Thiosemicarbazone FA4

**DOI:** 10.3390/molecules27051682

**Published:** 2022-03-04

**Authors:** Dario P. Anobile, Mauro Niso, Adrian Puerta, Stephanie M. Fraga Rodrigues, Francesca S. Abatematteo, Amir Avan, Carmen Abate, Chiara Riganti, Elisa Giovannetti

**Affiliations:** 1Department of Oncology, University of Torino, Via Santena 5/bis, 10026 Torino, Italy; dario.anobile@edu.unito.it; 2Department of Medical Oncology, Amsterdam UMC, VU University, Cancer Center Amsterdam, De Boelelaan 1117, 1081HV Amsterdam, The Netherlands; apuertaa@ull.edu.es (A.P.); s.m.fragarodrigues@amsterdamumc.nl (S.M.F.R.); 3Dipartimento di Farmacia-Scienze del Farmaco, Università degli Studi di Bari ALDO MORO, Via Orabona 4, 70125 Bari, Italy; mauro.niso@uniba.it (M.N.); francesca.abatematteo@uniba.it (F.S.A.); 4BioLab, Instituto Universitario de Bio-Orgánica “Antonio González” (IUBO-AG), Universidad de La Laguna, c/Astrofísico Francisco Sánchez 2, 38206 La Laguna, Spain; 5Basic Medical Sciences Institute, Mashhad University of Medical Science, Mashhad 91886-17871, Iran; avana@mums.ac.ir; 6Metabolic Syndrome Research Center, Mashhad University of Medical Science, Mashhad 91886-17871, Iran; 7Interdepartmental Research Center of Molecular Biotechnology, University of Torino, 10126 Torino, Italy; 8Fondazione Pisana per la Scienza, Via Ferruccio Giovannini 13, 56017 San Giuliano Terme, Italy

**Keywords:** multifunctional thiosemicarbazone, σ-2 receptor ligands, pancreatic cancer primary cultures, chemoresistance, migration

## Abstract

A new sigma-2 (σ2) receptor ligand (**FA4**) was efficiently synthesized and evaluated for cytotoxic, proapoptotic, and antimigratory activity on pancreatic ductal adenocarcinoma (PDAC) primary cell cultures, which restrained the aggressive and chemoresistant behavior of PDAC. This compound showed relevant antiproliferative activity with half maximal inhibitory concentration (IC50) values ranging from 0.701 to 0.825 μM. The cytotoxic activity was associated with induction of apoptosis, resulting in apoptotic indexes higher than those observed after exposure to a clinically relevant concentration of the gemcitabine, the first-line drug used against PDAC. Interestingly, **FA4** was also able to significantly inhibit the migration rate of both PDAC-1 and PDAC-2 cells in the scratch wound-healing assay. In conclusion, our results support further studies to improve the library of thiosemicarbazones targeting the σ-2 receptor for a deeper understanding of the relationship between the biological activity of these compounds and the development of more efficient anticancer compounds against PDAC.

## 1. Introduction

Pancreatic ductal adenocarcinoma (PDAC) represents about 95% of all types of pancreatic cancer and is characterized by an aggressive biological behavior and dismal prognosis. The global burden of PDAC has more than doubled over the past 25 years, and it is predicted to become the second cancer-related death reason in 2040 [1,2].

Five-year survival rates have just reached about 10%, but therapeutic options are limited, and often patients develop resistance to the standard treatments based on gemcitabine [3]. 

In the attempt to identify new vulnerabilities to overcome chemoresistance, the so-called sigma (σ-1 and σ-2) receptors have attracted special interest, because they are overexpressed in several tumors, including PDAC, compared to healthy tissues, and their ligands trigger cancer cell apoptosis by pleiotropic mechanisms, including intracellular calcium oscillation, endoplasmic reticulum stress, and alteration of mitochondrial polarization [4,5]. The σ1 receptor has been defined as a pluripotent modulator mainly residing in the mitochondria-associated ER membrane (MAM) where it exerts its functions through protein–protein interactions [6]. The σ2 receptor has only recently been identified as the TMEM97 protein. Its crystal structure has been resolved, helping in the understanding of the ligand–protein interaction and paving the way for more in-depth investigations of the protein biological functions [7,8]. Of note, 20-(S)-hydroxycholesterol has been identified as an endogenous interactor of σ2 receptor/TMEM97 [9].

Recently, Niso and collaborators have evaluated the anticancer activity of a series of σ2 receptor agonists belonging to the thiosemicarbazone family (including the **FA4** compound) in immortalized cell lines, both in vitro and in vivo. Of note, these ligands were also able to induce apoptosis in the human PDAC cell lines AspC1, MiaPaCa2, and PANC-1 in the low micromolar range [10]. σ-2 receptor agonists have been shown to induce apoptosis with limited toxicities in different preclinical models and, thus, are a promising possibility to trigger apoptosis or autophagy when chemoresistance to traditional therapies is developed [11,12].

Starting from this basis, our work aimed to further validate the effects of the better-performing thiosemicarbazone, namely **FA4**, in patient-derived primary cell cultures (PDAC-1 and PDAC-2), which are ex vivo cell populations recovered directly from fresh surgically resected samples from PDAC patients. These cells have the advantage of preserving the most important patient features because they are cultured in vitro for a few passages, in order to maintain the genetic hallmarks of the primary originator tumors [13]. Moreover, bioinformatic studies conducted in our laboratory highlighted that the gene coding for the σ-2 receptor was overexpressed in the PDAC-1 and PDAC-2 cells compared to the normal immortalized pancreatic ductal cells HPDE. These models were used to further establish the cytotoxic and proapoptotic effects of **FA4**, as well to investigate its inhibitory effects on cell migration.

Remarkably, we found that in both PDAC1 and PDAC2, the IC50 of the **FA4** compound was lower than the one reported in cancer immortalized cell lines, suggesting that the compound also exerts a good cytotoxic activity in primary cell cultures. This effect was associated with apoptosis induction, which was comparable to the proapoptotic effects of gemcitabine, a drug commonly used in the treatment of PDAC [14].

As for the second objective of our study, **FA4** has been shown to exert antimigratory effects and could, therefore, provide new opportunities for therapy against the pro-invasive and metastatic behavior of PDAC [15].

## 2. Results 

### 2.1. TMEM97 mRNA Expression in Pancreatic Cancer Primary Cells

Our next generation sequencing (NGS) transcriptomic data showed that the σ-2 receptor mRNA TMEM97 was expressed in both primary pancreatic cancer cell culture, at significantly higher values compared to the normal human pancreatic ductal epithelial HPDE cells (Figure 1). These results were in line with previous findings, demonstrating a higher expression of the σ2 receptors on the plasma membranes of PDAC cell lines compared to HPDE cells (Appendix A) [10]. Of note, the highest Fragments per Kilobase Million (FPKM) value was measured in PDAC-2 cells (Figure 1), which originated from the most clinically aggressive tumor [16].

### 2.2. ***FA4*** Synthesis 

**FA4** was obtained according to a polypharmacology strategy, following other σ2 receptor-targeting thiosemicarbazones (e.g., **MLP44** and **PS3**) that had shown promising antitumor properties in a panel of immortalized human PDAC cells and in a murine (KP02) tumor model [17,18]. All these thiosemicarbazones were designed with the aim to bind σ-2 receptors, that are overexpressed in a number of cancers [19,20], and chelate metals (in particular iron and copper ions) in order to obtain synergic effects by adding the effects of the alteration of the redox state of cells due to metal chelation to the cytotoxic properties proper of the σ2 agonists [21].

**FA4**, which was designed based on the σ2 reference agonist siramesine, demonstrated a superior anticancer activity in a panel of PDAC cells in comparison with its congeners **MLP44** and **PS3**, that only differ in the basic moiety, with the (*Z*)-*N*,*N*-dimethyl-2-(2-oxoindolin-3-ylidene)hydrazinecarbothioamide unchanged (Figure 2). 

The results obtained in commercial PDAC cell lines [10] highlighted how the basic moiety 3*H*-spiro[isobenzofuran-1,4′-piperidine] of **FA4**, which differs from the 6,7-dimethoxytetrahydroisoquinoline in **MLP44** or the 1-cyclohexylpiperazine in **PS3**, may drive this increased activity. While **MLP44** and **PS3** underwent a deconstruction approach in order to evaluate how the chelating moiety impacts on the σ receptor affinity and overall activity, this was not investigated for **FA4**. Indeed, the removal of the thiosemicarbazone moiety importantly reduced the cytotoxic activity of **MLP44** and **PS3** [10]. Thus, **FA4** was deconstructed by removing the thiosemicarbazone moiety, resulting in compound **MT8** that appears like a simplified isomer of the siramesine, in which the butyl linker is in position-1 rather than position-3 of the indole ring, devoid of the *p*-fluorophenyl portion.

According to a previously reported procedure [17], the synthesis of compound **MT8** was obtained by nucleophilic substitution of the spiropiperidine system on the 1-(4-chlorobutyl)-1*H*-indole portion (Figure 1).

The compounds underwent radioligand binding assays on σ1 and σ2 receptors, according to the previously adopted protocols [17,18]. While the affinity for the σ2 receptor was subnanomolar (*K*_i_ = 0.43 nM, Table 1), for **MT8,** the affinity for the σ-1 receptor was in the two-digit nanomolar range (*K*_i_ = 27.3 nM, Table 1): this compound was a moderately selective high-affinity σ-2 receptor ligand, with an improved profile compared to siramesine that, from our perspective, did not discriminate between the two σ subtypes (*K*_i_ = 10.5 nM for σ1 and *K*_i_ = 12.6 nM for σ2 receptors, respectively) [22]. 

These data also suggest how the thiosemicarbazone portion on this scaffold was slightly detrimental for the interaction with the σ2 binding site, although **FA4** retained an optimal binding profile in the nanomolar range (*K*_i_ = 15.8 nM for the σ1 and *K*_i_ = 51.3 nM for the σ2 receptors, respectively, Table 1). **MT8** was evaluated for its anticancer properties in the PANC-1 cell line, in which **FA4** activity appeared promising, both in vitro and in vivo [10]. **MT8** had an appreciable cytotoxic effect in the low micromolar range (IC_50_ = 19.1 μM), but it was six-fold less potent than **FA4**.

The cytotoxicity of **MT8** was also evaluated in other cancer cell lines, such as the lung adenocarcinoma A549 cells, the breast adenocarcinoma MCF7 cells, and the hepatocarcinoma HepG2 cells, displaying an IC_50_ = 4.29 μM in A549, IC_50_ = 6.02 μM in MCF7, and IC_50_ = 3.20 μM in HepG2 cells. These data highlight the higher aggressiveness of PANC-1 cells, in which **MT8** displayed the lowest potency. On the other hand, the data also confirmed the need for a strategy with a greater impact on pancreatic tumors. From this perspective, the multitarget **FA4** appeared to be a promising candidate and was selected to be further investigated in the primary PDAC cell cultures of the present study.

### 2.3. Antiproliferative Activity 

Cell viability upon exposure to **FA4** was measured with the Sulforhodamine B (SRB) assay. **FA4** caused a concentration-dependent inhibition of proliferation (Figure 3), with IC50 values of 0.825 ± 0.006 and 0.701 ± 0.059 μM in the PDAC-1 and PDAC-2 cells, respectively (Table 2). Lastly, we performed additional experiments to evaluate the in vitro cytotoxicity of **FA4** against normal HPDE cells and normal fibroblast Hs27 cells. The results of these studies allowed us to calculate the selectivity index (SI, calculated as IC50 non-tumor cells/IC50 tumor cells), which were 11 and 7 in HPDE and Hs27 cells, respectively. Interestingly, **FA4** had SI values similar to the values measured for gemcitabine, as reported in previous studies in the same PDAC primary cells [23]. 

Of note, the IC50 values of **FA4** in PDAC-1 and PDAC-2 cells were lower than the values reported in human commercial PDAC cell lines, namely BxPC3, AspC1, MiaPaCa2, and PANC-1, where they ranged between 0.88 μM and 3.01 μM [10].

IC50 values ± SD were determined by the graphical interpolation of the dose–response curves reported in Figure 3.

### 2.4. Apoptosis Induction

Previous studies reported that the modulation of σ2 receptor activity contributes to the initiation of the apoptotic process [4,10]. Therefore, we evaluated the effect of **FA4** on the induction of apoptosis in the PDAC primary cultures. To this aim, we measured the externalization of the plasma membrane phosphatidylserine, a reliable marker of cell apoptosis, which was quantified by the measurement of the fluorescence of annexin-V by spectrophotometric assay and microscopy. After 24 h of treatment with 10 μM **FA4**, a significant increase in the portion of apoptotic cells was observed compared to untreated control cells. The highest apoptotic index was measured in the PDAC-2 cells (17.6 ± 3.6, compared to 15.5 ± 4.9 in the PDAC-1 cells).

Remarkably, the percentages of apoptotic cells after treatment with **FA4** were significantly higher compared to the number of cells that underwent apoptosis after treatment with gemcitabine at 10 μM (Figure 4), indicating that **FA4** had superior anticancer activity. 

### 2.5. Migration Assay 

Cancer cell migration is a key factor contributing to the aggressiveness of PDAC. It is therefore important to find new compounds that can counteract or stop this process [24]. To evaluate the effect of **FA4** on cell migratory properties, a migration assay was performed by incubating cells for 24 h with **FA4**. As reported in Figure 5 and Appendix A, the compound was able to inhibit migration over 24 h with respect to the control and had an antimigratory activity superior to previous data with gemcitabine in these PDAC cell cultures [25,26]. 

## 3. Discussion

PDAC is a highly aggressive tumor, and therapeutic options are limited by constitutive and acquired resistance to treatment. In the search for a novel way to improve this dismal scenario, σ2 receptors have recently demonstrated encouraging results both in vitro and in vivo, showing the efficacy of novel compounds such as **FA4** especially in PDAC cells resistant to gemcitabine [10]. To further define the potentiality of this promising drug, we studied the activity of **FA4** in primary patient-derived cell cultures. These PDAC preclinical models, mimicking the molecular complexity of the original PDAC tumors, represent an improvement for the experimental testing of anticancer agents and might reduce their failure in follow-up clinical trials [16].

First, we found that the σ2 receptor/TMEM97 gene was significantly upregulated compared to normal pancreatic epithelial cells, particularly in patient-derived cell lines with particular aggressiveness [16]. This observation prompted us to investigate how σ2 receptor-targeting agents performed compared to gemcitabine, currently used as first-line therapy, with a special focus on advanced and chemorefractory tumors, where the efficacy of gemcitabine is low [3]. 

Following a multi-targeting drug synthesis approach, we designed a series of σ2 receptor agonists [21]. A preliminary screening on different commercial PDAC cell lines, indicated **FA4**, based on the modification of the structure of σ2 reference agonist siramesine, as the lead compound in terms of cytotoxicity [10]. By further deconstructing the siramesine structure, we produced a second congener, **MT8**, devoid of the thiosemicarbazone portion. Although **MT8** had a higher affinity for the σ2 receptor than **FA4**, its cytotoxic profile was worse in the PANC-1 cell line, while it showed a good cytotoxic profile (in the same range as **FA4**) in other cancer cell lines. This result suggests that the σ-2 engagement by **MT8** could be responsible for part of the cytotoxicity effects, but that the presence of the thiosemicarbazone moiety in **FA4** importantly improved its cytotoxicity in PDAC cells. These data are of particular interest because they were obtained in PANC-1 cells, which are typically resistant to the activity of gemcitabine [27].

Prompted by these results, we narrowed our focus to **FA4** and compared its biological effects with those of gemcitabine in primary PDAC samples. First, dose–response assays to measure cell viability indicated a good antitumor effect on PDAC cell and a higher IC50 on non-transformed cells, such as pancreatic epithelial cells and fibroblasts. Second, the SI was comparable for **FA4** and gemcitabine [23]. Together, these results indicated that **FA4** has a potentially good therapeutic window.

Since σ2 receptor activity is known to trigger apoptosis in PDAC cells [4,10], we next tested the proapoptotic effect of **FA4** in our primary PDAC cells. Interestingly, **FA4** induced a higher percentage of apoptotic cells than gemcitabine. We considered that **FA4** had stronger anticancer activity but the same SI of gemcitabine; hence, our results indicate that **FA4** was a stronger and safer anti-PDAC agent, at least in vitro. **FA4** was also safe in PDAC xenografts [10], suggesting that it could be noteworthy of further investigation in patient-derived xenografts (PDXs) and in a clinical setting. Furthermore, **FA4** also decreased the migration of PDAC cells, revealing another important property in the treatment of PDAC, a cancer known for its high invasiveness [24].

These results prompt further studies on the multiple effects of **FA4** on primary patient-derived cell cultures and PDXs and on the pharmacological effects of a co-incubation of **FA4** with the drugs that represent the actual standard of care for PDAC, such as gemcitabine. Other future work will help to better clarify the pharmacokinetics and pharmacodynamics of what we hope will be a new drug in the fight against such aggressive tumor.

## 4. Materials and Methods

### 4.1. Cell Culture

Human pancreatic adenocarcinoma primary cultures (PDAC1 and PDAC2) were cultured in RPMI + 10 % New Born Calf Serum + 1% Penicillin/ Streptomycin and maintained in standard conditions for <20 passages. Cells were routinely tested for *Mycoplasma* spp. and were free of contamination. 

Human pancreatic duct epithelial-like cell line hTERT-HPNE (CRL-4023™) and human skin fibroblasts Hs27 (CRL-1634™), obtained from the American Type Culture Collection (ATCC^®^, Manassas, VA, USA), were cultured in supplemented KGM medium (Lonza) and DMEM media, respectively, with 10% heat inactivated Fetal Bovine Serum (FBS) and 1% penicillin (100 mg/mL) and streptomycin (100 mg/mL). 

Previous experiments were performed with human pancreas cancer cell lines PANC-1 (CRL-1469™), MiaPaCa2 (CRL-1420™), BxPC-3 (CRL-1687™), PANC-02.03 (CRL-2553™), AsPC-1 (CRL-1682™), human A549 lung cancer cells (CRM-CCL-185™), human MCF7 adenocarcinoma breast cells (HTB-22™), and human HepG2 hepatocarcinoma cells (HB-8065™), obtained from ATCC. PANC-1, MiaPaCa2, BxPC-3, PANC-02.03, AsPC-1, and MCF7cells were cultured in DMEM with heat inactivated 10% FBS, penicillin (100 mg/mL), and streptomycin (100 mg/mL). HepG2 cells were cultured in MEM supplemented with 10% (*v*/*v*) heat inactivated FPS, 1% (*v*/*v*) glutamine, 1% (*v*/*v*) penicillin, and streptomycin, 1% (*v*/*v*) NEAA. A549 cells were grown in HAM’S F12 supplemented with 10% heat inactivated FBS, 2 mM glutamine, 100 U mL^−1^ penicillin, and 100 μg/mL streptomycin. All these cells were kept at 37 °C in a humidified incubator under an atmosphere of 5% CO_2_ in 75 cm^2^ tissue culture flasks (Greiner Bio-One GmbH, Frickenhausen, Germany) and were harvested with trypsin-EDTA in their exponentially growing phase.

### 4.2. Evaluation of the mRNA Expression of the Gene Coding for σ-2 Receptor (TMEM97) 

RNA-sequencing analyses for PDAC-1 and PDAC-2 were performed, as described by Sciarrillo and collaborators [28]. The raw data were preprocessed for quality filtering and adapter trimming using the FASTX Toolkit (version 0.7) and subsequently mapped to the GRCh38 version of the human genome using the STAR alignment tool (version 2.5.3a). With these methods, we obtained around 90% of reads mapped to the human genome per sample. Gene counts in Fragments per Kilobase of transcript per Million mapped reads (FPKM) normalization were computed using the CuffLinks algorithm.

### 4.3. σ2 Receptor Measurement by Flow Cytometry

Here, 1 × 10^6^ HPDE, MiaPaCa2, PANC-1, BxPC-3, PANC-02.03, and AsPC-1 cells were incubated for 75 min at 37 °C with 100 nM of the σ2 fluorescent ligand 2-{6-[2-(3-(6,7-dimethoxy-3,4-dihydroisoquinolin-2(1*H*)-yl)propyl)-3,4-dihydroisoquinolin-1(2*H*)-one-5-yloxy]hexyl}-5-(dimethylamino)isoindoline-1,3-dione (NO1) that completely saturated the σ2 receptors in these experimental conditions [29]. Then, 10 μM (+)-pentazocine was co-incubated to mask the σ1 receptors and detect the σ2 receptor specific binding. After washing with PBS, cells were detached with 0.2 mL Cell Dissociation Solution (Sigma-Aldrich-RBI, Milan, Italy)), centrifuged at 13,000× *g* for 5 min, and rinsed with 0.5 mL PBS. The fluorescence, an index of the σ2 receptor amount on the plasma-membrane [29], was measured on 5 × 10^5^ cells, using a Guava Easy-Cyte™ 5 Flow Cytometry System (Millipore, Billerica, MA, USA), with a 530 nm band pass filter. The results were analyzed using the InCyte software (Millipore).

### 4.4. Chemical Synthesis and Analysis

Column chromatography was performed on ICN silica gel 60 Å (63–200 μm) as the stationary phase. Mass spectra were recorded with a HP GC/MS 6890-5973 MSD spectrometer, electron impact 70 eV, equipped with HP ChemStation. Then, ^1^H-NMR spectra were recorded in CDCl_3_ on a 500-vnmrs500 Agilent spectrometer (499,801 MHz). Analytical HPLC was performed on an Agilent 1260 Infinity Binary LC System equipped with a diode array detector using a reversed phase column (Phenomenex Gemini C-18, 5 mm, 250 × 4.6 mm).

To synthesize 1’-(4-(1*H*-indol-1-yl)butyl)-3*H*-spiro[isobenzofuran-1,4’-piperidine] (**MT8**), a mixture of 1-(4-chlorobutyl)-1*H*-indole (100 mg, 0.483 mmol), 3*H*-spiro[isobenzofuran-1,4’-piperidine] (109 mg, 0.58 mmol), and K_2_CO_3_ (80 mg, 0.58 mmol) in CH_3_CN (8 mL) [17,30] was heated at reflux overnight. After cooling, the solvent was removed under reduced pressure, and water was added to the residue. The mixture was extracted with CH_2_Cl_2_ (2 × 10 mL) and EtOAc (2 × 10 mL). The collected organic layers were dried (Na_2_SO_4_) and evaporated to afford a residue, which was purified by column chromatography (1:30) with CH_2_Cl_2_/MeOH (95:5) as eluent to give the final compound as a yellow oil (30 mg, 19% yield), which was converted into the hydrochloride salt through gaseous HCl in Et2O. GC-MS, and 1H-NMR spectra were collected on the free base. GC-MS *m*/*z*: 360 (M+, 9), 202 (100), 130 (12); ^1^H-NMR (500 MHz, CDCl3) δ: 1.55–1.65 (m, 2H, NCH_2_C*H*_2_), 1.76 (dd*, 2H, *J*1 = 12 Hz, C(C*H*H)2), 1.85–1.95 (m, 2H, NCH_2_CH_2_C*H*_2_), 1.99 (dt, 2H, *J*1 = 12 Hz, *J*2 = 4.4 Hz, C(CH*H*)_2_), 2.35–2.45 (m, 4H, (C*H*H)_2_NC*H*_2_), 2.80–2.84 (m, 2H, (CH*H*)_2_N) 4.17 (t, 2H, *J* = 7.3 Hz, NCH_2_), 5.05 (s, 2H, OCH_2_Ar), 6.49 (d, 1H, *J* = 2.9 Hz, aromatic), 7.08–7.29 (m, 7H, aromatic), 7.37 (d, 1H, *J* = 7.82 Hz, aromatic), 7.63 (d, 1H, *J* = 7.82 Hz, aromatic). **J*2 could not be calculated from the spectrum as it was too small. Hydrochloride salt was >98 % pure by HPLC analysis performed by isocratic elution with CH_3_CN/HCOONH_4_ (20 mM, pH = 5) 80:20 *v*/*v*, at a flowrate of 1 mL/min.

### 4.5. Radioligand Binding Assays

*σ*1 and *σ*2 receptor binding were carried out following Matsumoto et al. [31]. [^3^H]-DTG (30 Ci/mmol) and (+)-[^3^H]-pentazocine (34 Ci/mmol) were purchased from PerkinElmer Life Sciences (Zavantem, Belgium). DTG was purchased from Tocris Cookson Ltd. (Bristol, UK). (+)-Pentazocine was obtained from Sigma-Aldrich-RBI. All the procedures for the binding assays were described previously by Abate and collaborators [32].

### 4.6. Sulforhodamine-B (SRB) Assay

Cells were seeded in 96-well flat-bottom plates at a density of 3000/well, using both a control plate and an experimental plate. After 24 h, the control plate was fixed with cold 50% trichloroacetic acid, and the incubation with the compound of interest was performed in the experimental plate. After 72 h of incubation, the experimental plate was also fixed. Subsequently, both plates were washed and stained with the SRB solution, and after vigorous washing and drying, the bound dye was extracted from the cells with an alkaline solution. The absorption of SRB was measured spectrophotometrically at a wavelength of 490 nm and 540 nm. The intensity of the signal is proportional to the number of living cells and, therefore, a measure of their proliferation. Thus, cell growth inhibition was calculated as the percentage versus vehicle-treated cells (“negative control”) OD (corrected for OD of the control plate before drug addiction). Finally, the half maximal inhibitory concentration (IC50) was calculated by nonlinear least squares curve fitting (GraphPad Prism 7, Intuitive Software for Science, San Diego, CA, USA), as described previously [33]. The SRB assay was performed in technical and biological triplicates.

### 4.7. Apoptosis Assay

Cells were seeded in 96-well plates (5 × 10^3^ cells/well), and after 24 h, the cells were fixed with 4% paraformaldehyde in PBS for 30 min, then washed twice with PBS and stained with annexinV-FITC in binding buffer (10 mM HEPES/NaOH pH 7.4, 140 mM NaCl, and 2.5 mM CaCl_2_) for 10 min in the dark at room temperature. Lastly, cells were washed with the assay binding buffer solution, and fluorescence was measured by using a multimode plate reader with excitation and emission filters at 485 nm and 535 nm, respectively, to determine the apoptotic index. The values were normalized considering the cell proliferation as assessed by crystal violet assay, as described previously [34]. Cells were also stained with 8 μg/mL bisbenzimide to detect typical apoptotic morphological features, as described previously [35].

### 4.8. Migration Assay

Cells were seeded into 96-well plates 72 h before the assay at a density of 5 × 10^4^ /well and allowed to attach and to become confluent. Subsequently, a scratch was performed in every well with a 96-pin scratcher, allowing the formation of uniform scratches. Immediately after this passage, cells were cultured for 24 h in complete medium with the drug of interest for experimental wells or without any drug for the control wells. Pictures were taken with an optical microscope at different time points (0, 6, and 24 h) and analyzed with the Scratch Assay 6.2 software (Digital Cell Imaging Labs, Keerbergen, Belgium). The migration assay was performed in technical and biological triplicates, as reported previously [35].

### 4.9. Statistical Analysis

All the assays were carried out in triplicate and repeated at least three times, whereas the percentages of apoptotic cells were calculated taking into account at least three biological replicates. The data were evaluated using GraphPad Prism 9.0 (GraphPad Software, San Diego, CA, USA). Data were analyzed by applying the one-way repeated measures analysis of variance and Bonferroni’s multiple comparison test as a post hoc test. The results were reported as mean ± SD (standard deviation), and statistical significance was accepted at *p* < 0.05.

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
