# Peer review of "New Pharmacological Strategies against Pancreatic Adenocarcinoma: The Multifunctional Thiosemicarbazone FA4"

_molecules, 2022, doi:10.3390/molecules27051682_

Round 1

Reviewer 1 Report

the manuscript is very interesting considering the relevance of prostate cancer and the life-threatening type of cancer. however, some concerns arose which are listed below:

  • figure 1: the expression levels done by PCR should be supported by Western blot analysis. If possible, the plasma membrane localization should be also proven to strengthen the relevance of the proposed results.
  • table 1: it is not clear why for siramesine and pentazocine no cytotxicity is tested. furthermore, does DTG  not bind to sigma1 receptor? The flow cytometry assay referenced in numb4 should be explained in the manuscript considering that is the method employed to collect the proposed results
  • paragraphs 2.3, 2.4 and 2.5: why is the MT8 compound not tested for antiproliferative effects, apoptosis induction, and migration assay?
  • figure 3b: IC50 curves should be shown and IC50 values reported in a table. this graph is not informative in this version.
  • docking analysis of the compounds with 3D homology models of the receptor would much improve the manuscript. The receptor structures are now available in alphafold database (https://alphafold.ebi.ac.uk/) if the protein structures are not known yet.

Author Response

Reply to Reviewer 1

Comments and Suggestions for Authors

The manuscript is very interesting considering the relevance of prostate cancer and the life-threatening type of cancer. however, some concerns arose which are listed below:

1) Figure 1: the expression levels done by PCR should be supported by Western blot analysis. If possible, the plasma membrane localization should be also proven to strengthen the relevance of the proposed results. …

We already measured the amount of plasma-membrane associated σ-2 receptor in 5 different PDCA cell lines and in HPDE cells. The results, reported in [Niso M et al, Cell. Oncol. 2021 44, 1307–1323, ref. 10 of the new version], are presented as Supplementary Figure S1 and show that PDAC cell lines have higher levels of σ-2 receptor than non-transformed HPDE cells. We could not perform the same experiments on primary cells because of the low amount of biological materials, but we believe that our panel of commercially available PDAC cells, with different genetic background and aggressiveness, recapitulates the phenotype observed in PDAC cells compared to normal pancreatic epithelial cells. We modified the Results (line 92), the Discussion (line 252) and the Materials and Methods (line 327) sections accordingly.

2) Table 1: it is not clear why for siramesine and pentazocine no cytotxicity is tested. furthermore, does DTG  not bind to sigma1 receptor? The flow cytometry assay referenced in numb4 should be explained in the manuscript considering that is the method employed to collect the proposed results.

We apologize because we inserted the wrong Table in the text. This revised version of the manuscript displays the correct one. For both siramesine and pentazocine, the IC50 was > 50 µM. This lack of cytotoxicity is in line with a recent paper showing that siramesine has no effect on PDAC tumor size [Cash TP et al, Cancers 2020 12, 1790]. (+)-Pentazocine is a σ-1 agonist and thus it is devoid of the cytotoxicity that characterizes instead the σ-2 agonists.

DTG is a σ-2 receptor unselective agonist, indeed it also binds σ-1 receptor with a similar affinity, as reported in Table 1. Hence, DTG is not considered a reference compound for σ-2 receptor-induced cytotoxicity. Indeed, the cytotoxicity of DTG is low. The value reported in the previous version was a typo, we apologize for the mistake. We were not able to determine a precise IC50 for siramesine, pentazocine and DTG because the compounds loose solubility at higher concentration (they become only soluble in methanol, that is cytotoxic as solvent alone).

As indicated in Table 1, only the affinity of FA4 for the σ-1 subtype has been evaluated by flow cytometry, while traditional radioligand binding protocols on animal tissues were used for the other binding values at σ receptors (paragraph 3.2 of the Materials and Methods section, reference 22 of the new version). The data about FA4 binding at σ-1 and the detailed methods were already reported in or recently published paper [Niso M et al, Cell. Oncol. 2021 44, 1307–1323, ref. 10 of the new version]. We do not consider appropriate to re-copy them in this new manuscript, since the interested reader may easily access the flow cytometry protocols referring to the above paper.

For this Reviewer convenience, we report the protocol below:

“FA4 binding to σ-1 in PANC-1, MiaPaCa2 and HPDE cells that were incubated with increasing concentrations (1, 10 and 100 nmol/L and 1 and 10 μM) of (+)-pentazocine or FA4, followed by 100 nmol/L sigma-1 fluorescent compound (LM1, 5-(dimethylamino)-2-(6-((5-(4-(4-methylpiperidin-1-yl)butyl)-5,6,7,8-tetrahydronaphthalen-2-yl)oxy)hexyl) isoindoline-1,3-dione) [14] for 75 min at 37 °C. To mask σ-2 receptors, the σ-2 receptor selective ligand F390, 2-(3-(6,7-dimethoxy-3,4-dihydroisoquinolin-2(1H)-yl)propyl)-5-methoxy-3,4-dihydroisoquinolin-1(2 H)-one (10 μM) was co-incubated. At the end of the incubation periods, the cells were washed twice with PBS, detached with 200 ml Cell Dissociation Solution (Sigma Chemical Co.) for 10 min at 37 °C, centrifuged at 13,000 g for 5 min and resuspended in 500 μl PBS. Fluorescence was recorded using a Bio-Guava® easyCyte™ 5 Flow Cytometry System (Millipore, Billerica, MA, USA), equipped with a 530 nm band pass filter. For each analysis, 50,000 events were collected and analyzed using InCyte software (Millipore).”

3) Paragraphs 2.3, 2.4 and 2.5: why is the MT8 compound not tested for antiproliferative effects, apoptosis induction, and migration assay?

As stated in the text (line 164), MT8 displayed cytotoxicity in the low micromolar range in lung, breast and liver cancer cell lines, but it was 6-fold less cytotoxic than FA4 in the PANC-1 cell line. We believe that the cytotoxicity potential of MT8 could be explored tumors other than PDAC, while FA4 is the lead compound in terms of cytotoxicity for this aggressive cancer. For this reason, we focused only on FA4 for the subsequent experiments on primary PDAC cells. We better specified this point in the Results (line 170) and in the Discussion (line 261).  

4) Figure 3b: IC50 curves should be shown and IC50 values reported in a table. this graph is not informative in this version.

As requested, we reported the whole dose-response curves for PDAC1, PDAC2 and two normal cell lines (human pancreatic epithelial HPDE cells and human fibroblasts Hs27 cells) in the new Figure 3. The IC50 value were reported in the new Table 2. We modified the text accordingly (line 177).

5) Docking analysis of the compounds with 3D homology models of the receptor would much improve the manuscript. The receptor structures are now available in alphafold database (https://alphafold.ebi.ac.uk/) if the protein structures are not known yet.

We thank the Reviewer for the suggestion. We believe that the docking study is out of the focus of the present work that is centered on the biological properties of FA4 against patient derived PDAC cells. In an ongoing project, we are working on the further optimization of FA4. The starting point of this project is the docking of FA4 with the receptor. The project is ongoing, and we are not able to provide the data now. These results will be included in a future work.

Reviewer 2 Report

The reviewed article shows thiosemicarbazone FA4 as an anticancer agent and its possible application in the treatment of pancreatic adenocarcinoma. The pharmacological properties of the tested compound as well as its mechanism of action seem to be interesting and valuable. The manuscript is written correctly but too general. Presented results of in vitro studies only allow general conclusions to be drawn. They do not indicate specific mechanisms of action. The issues that need to be clarified or corrected are below:

  • The Abstract subsection includes the following sentence: “The antiproliferative activity was associated with induction of apoptosis, resulting in apoptotic indexes…”. Please consider the term "antiproliferative effect" as a cytostatic effect and "cytotoxic effect" leading to cell damage and induction of apoptosis. In my opinion, the sentence is imprecise.
  • The Introduction section: Please add new and more detailed information about: i) epidemiological data of pancreatic ductal adenocarcinoma (e.g. morbidity in recent years); ii) sigma receptors, especially sigma 2 (molecular structure, subcellular localization, cytophysiological function).
  • In the introduction, please specify the nature of the ligand (agonist).
  • I strongly recommend editing the manuscript and splitting the results and discussions into 2 separate sections.
  • SEM should be replaced by SD. Please recalculate the results and statistics and make corrections to the graphs.
  • The caption of Scheme 1 is confusing - it presents MT8 synthesis.
  • Table 1 - The result of IC50 for FA4 is without SEM (SD).
  • Figure 3a – Please present graphs for other cell lines. Moreover, I suggest presenting calculated values of IC50 in a table or under the corresponding graph.
  • Figure 5 – Please present representative pictures also for the PDAC-1 cell line.
  • Did the authors perform a microscopic evaluation of the cells? Microscopic photography showing the morphology of cells would certainly improve the reliability of the work.
  • Why does subsection 3.1 contain information about previously tested cell lines?

Author Response

Reply to Reviewer 2

Comments and Suggestions for Authors

The reviewed article shows thiosemicarbazone FA4 as an anticancer agent and its possible application in the treatment of pancreatic adenocarcinoma. The pharmacological properties of the tested compound as well as its mechanism of action seem to be interesting and valuable. The manuscript is written correctly but too general. Presented results of in vitro studies only allow general conclusions to be drawn. They do not indicate specific mechanisms of action. The issues that need to be clarified or corrected are below:

1) The Abstract subsection includes the following sentence: “The antiproliferative activity was associated with induction of apoptosis, resulting in apoptotic indexes…”. Please consider the term "antiproliferative effect" as a cytostatic effect and "cytotoxic effect" leading to cell damage and induction of apoptosis. In my opinion, the sentence is imprecise.

We thank the Reviewer for the criticism. We modified the Abstract accordingly (line 28).

2) The Introduction section: Please add new and more detailed information about: i) epidemiological data of pancreatic ductal adenocarcinoma (e.g. morbidity in recent years); ii) sigma receptors, especially sigma 2 (molecular structure, subcellular localization, cytophysiological function)

As suggested by this Reviewer, we added the epidemiological data on PDAC (line 42) and more detailed information about σ receptors biology (line 52) in the Introduction. We added 6 new references.

3) In the introduction, please specify the nature of the ligand (agonist).he term agonist was added to define FA4, as suggested (lines 60 and 63). We also added one new reference.

4) I strongly recommend editing the manuscript and splitting the results and discussions into 2 separate sections.

As requested, we split the Results and the Discussion into two separate sections in the revised version.

5) SEM should be replaced by SD. Please recalculate the results and statistics and make corrections to the graphs.

As requested, we replaced SEM with SD. We recalculated the statistics in all the graphs and we modified the Figure legends and the Material and Methods section (line 422).

6) The caption of Scheme 1 is confusing - it presents MT8 synthesis.

We thank this Reviewer for pointing it out. We apologize for the mistake that we fixed accordingly

7) Table 1 - The result of IC50 for FA4 is without SEM (SD).

We added the SD to the IC50 of Table 1.

8) Figure 3a – Please present graphs for other cell lines. Moreover, I suggest presenting calculated values of IC50 in a table or under the corresponding graph.

As requested, we reported the whole dose-response curves for PDAC1, PDAC2 and two normal cell lines (human pancreatic epithelial HPDE cells and human fibroblasts Hs27 cells) in the new Figure 3. The IC50 value were reported in the new Table 2. We modified the text accordingly (line 177).

9) Figure 5 – Please present representative pictures also for the PDAC-1 cell line.

As requested, we included representative pictures of the PDAC1 cell line as Supplemental File 2. 

10) Did the authors perform a microscopic evaluation of the cells? Microscopic photography showing the morphology of cells would certainly improve the reliability of the work.

We thank the Reviewer for the observation. We included the microscope-based evaluation of the cell morphology as new Figure 5c. We modified the Figure legend and the Materials and methods accordingly (line 401).

11) Why does subsection 3.1 contain information about previously tested cell lines?

We reported the information about cell cultures of other cell lines because in the initial evaluation of the cytotoxic properties of MT8 and FA4, we tested the effects on cell viability in human A549 lung cancer cells, human MCF7 adenocarcinoma breast cells and human HepG2 hepatocarcinoma cells. The results referred to these cell lines were reported in the text (Results, line 165). The comparison with the results obtained in PANC-1 cell line helped in identifying FA4 as the lead compound in terms of anti-cancer effects against PDAC.

Round 2

Reviewer 1 Report

authors addressed my requests

Author Response

We thank the Reviewer for the positive comments. 

Reviewer 2 Report

In response to the suggestions, the Authors improved the manuscript. They also answered all questions. However, one issue has not been corrected. Scheme 1 still presents MT8 synthesis and the caption is as follow: "Schematic representation of the FA4 synthesis"

Author Response

We thank the Reviewer for the positive comments. We corrected the mistake in the legend of Scheme 1.